# Subjective Evaluation of Text-to-Speech Models: Comparing Absolute Category Rating and Ranking by Elimination Tests

*Kishor Kayyar Lakshminarayana[1], Christian Dittmar[1], Nicola Pia[1], Emanuël A.P. Habets[2]*

[1]Fraunhofer Institute for Integrated Circuits (IIS), Am Wolfsmantel 33, 91058 Erlangen, Germany
[2]International Audio Laboratories Erlangen*, Am Wolfsmantel 33, 91058 Erlangen, Germany

{kishor.kayyar.lakshminarayana, christian.dittmar, nicola.pia}@iis.fraunhofer.de,
emanuel.habets@audiolabs-erlangen.de

## Abstract

Modern text-to-speech (TTS) models are typically subjectively evaluated using an Absolute Category Rating (ACR) method. This method uses the mean opinion score to rate each model under test. However, if the models are perceptually too similar, assigning absolute ratings to stimuli might be difficult and prone to subjective preference errors. Pairwise comparison tests offer relative comparison and capture some of the subtle differences between the stimuli better. However, pairwise comparisons take more time as the number of tests increases exponentially with the number of models. Alternatively, a ranking-by-elimination (RBE) test can assess multiple models with similar benefits as pairwise comparisons for subtle differences across models without the time penalty. We compared the ACR and RBE tests for TTS evaluation in a controlled experiment. We found that the obtained results were statistically similar even in the presence of perceptually close TTS models.

## 1. Introduction

Many modern text-to-speech (TTS) synthesis models [1, 2, 3] achieve nearly natural quality speech. This naturalness in quality is typically measured through subjective Absolute Category Rating (ACR) tests, which provide a Mean Opinion Score (MOS) per model. In a recent study, Cooper et al. [4] reported that the best five neural TTS models all had comparable quality. This implies a need to conduct subjective tests that differentiate between the quality of such TTS models.

In an ACR test, the outputs from TTS models (conditions) to be evaluated are presented one after the other. This has the disadvantage that the listener cannot go back and forth between the conditions to assess them. The multiple stimuli with hidden reference and anchor (MUSHRA) test [5] avoids this issue by presenting multiple conditions at the same time to the listener, e.g., [6]. Such a test is difficult to conduct in a TTS scenario since the 3.5 kHz or 7 kHz anchor signals used typically in such tests may not qualify as good anchor signals, e.g., [7]. The utterance's duration and prosody vary across synthesis models, and hence are unsuited for a MUSHRA test. Also, it might not be possible to use ground truth signal references. Further, [8] found that MUSHRA and MOS scores were comparable, but the difference between the TTS models was more apparent in MUSHRA.

Wester et al. [9] reported that at least 30 listeners and 150 total judgments per MOS value are required for reliable results with the ACR test. It would be interesting to know if there is an objective test that can show reliable results with less number of listeners.

When the difference between the samples under evaluation are subtle, providing absolute ratings to the samples becomes difficult and subjective. Pairwise comparisons can avoid this issue by providing for absolute comparison between the models. [10].These have also been used for subjective evaluation with TTS. The number of pairwise comparisons increases exponentially with the number of TTS models to be tested, thereby increasing the time and costs required for such an evaluation.

In the audio coding community, ranking-by-elimination (RBE) has been used successfully in the assessment across multiple models instead of a pairwise comparison test. In the RBE test, the evaluated samples are presented in a multi-stimulus fashion. The listener is expected to repeatedly eliminate the worst among the given samples until they cannot distinguish between the remaining items. The RBE is an indirect scaling test without absolute scores and hence easier to perform [10]. Wickelmaier et al. [11] compared the pairwise comparison test to the RBE test for audio codec evaluation. They showed that the results of pairwise comparison and the RBE test were comparable, but the RBE test was a lot faster. For example, the RBE test took 50 seconds to evaluate a particular audio sample, whereas the pairwise comparison took 300 seconds.

The current work compares the RBE test to the ACR test for speech synthesis evaluation. We simulate synthesis models with outputs at different perceptual qualities using the intermediate training checkpoints of a ForwardTacotron [3] model. Typically the quality of the model output improves progressively with training, which means that the expected quality changes are known. e.g., we know that the model trained for five epochs is worse than the model trained for 30 epochs. Since we are comparing evaluation methods, this prior knowledge is beneficial. We further show that the ACR and the RBE tests can be employed reliably to evaluate TTS models even when the outputs are perceptually very hard to distinguish. Both tests require a similar number of ratings for reliability. There are multiple open-source software tools available for subjective testing of audio like WebMUSHRA [12] and Hulti-Gen-v2 [13], but they do not support RBE test. Hence as an additional contribution, we implemented code to perform the RBE test as an extension to the WebMUSHRA software. This extension has been open-sourced for easy adoption and use in the community.

## 2. Evaluation Methods Under Test

### 2.1. Absolute category rating test

Typically subjective listening tests for evaluation of TTS models are done through absolute category rating tests. The mean of the rated categories across the obtained ratings is then reported as the MOS following the ITU-P.808 recommendation [14]. A five-point category-judgment scale is used for this purpose.

---

*A joint institution of the Friedrich-Alexander-Universität Erlangen-Nürnberg (FAU) and Fraunhofer IIS, Germany.

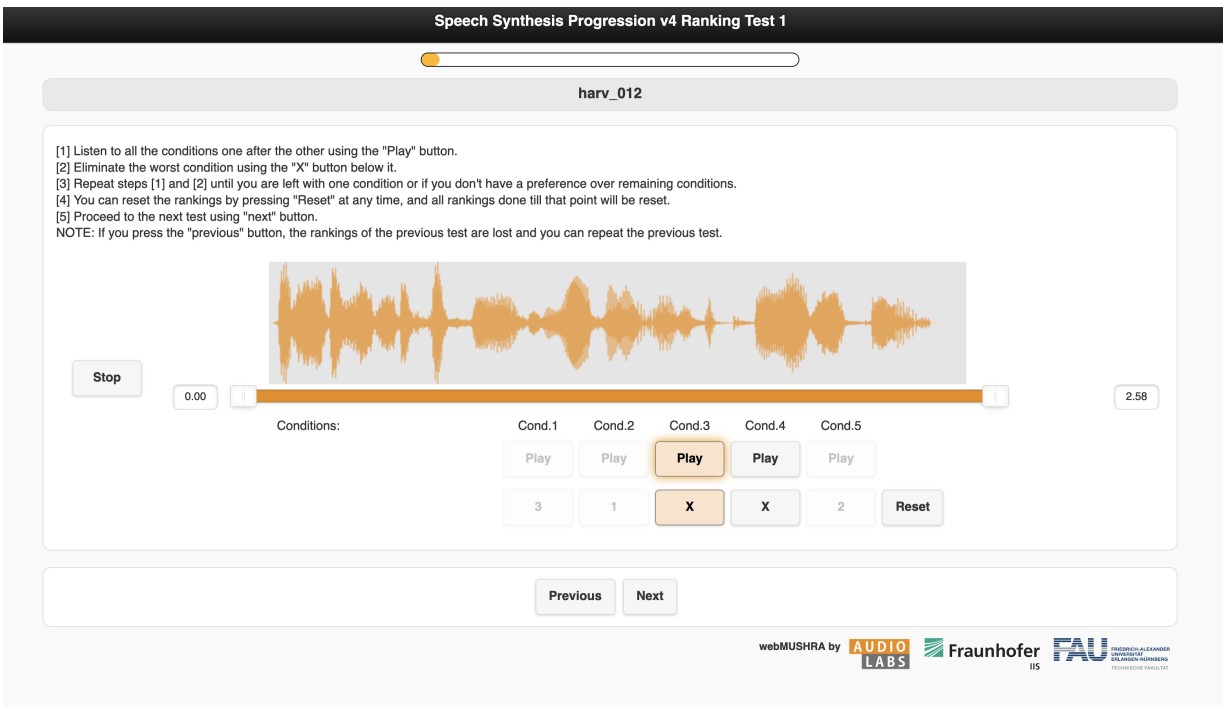

Figure 1: *The screenshot of the ranking-by-elimination test for one sample sentence. The eliminated samples are hidden, with the order of their elimination visible as their ranks. A lower rank indicates the sample was eliminated earlier, with lower quality.*

Each test sentence should be synthesized using all models under test and presented one after the other in a randomized order. The sentences used in the test should also be presented at random to the listener. The listener is expected to rate the presented sample using the ACR scale. The test can, for example, be conducted using the WebMUSHRA [12] software with a Likert-single-stimulus scale.

The ACR test consists of training and test parts, which are expected to be performed consecutively. Both parts are to be conducted similarly, but the training ratings are not included in the overall score. During the training, the listeners are expected to familiarize themselves with the test setup and are introduced to the typical artifacts in the samples. Further, the listeners are asked to use good-quality headphones and not change the volume once the test part begins.

The resultant ratings across all the listeners and conditions are averaged, and the confidence intervals are calculated. Further, the Wilcoxon-signed-rank tests are used to evaluate the statistical significance.

### 2.2. Ranking-by-elimination test

The RBE test is a multi-stimulus test, wherein the test sentence synthesized through each of the models under test is presented simultaneously to the listener [11]. The TTS models used to synthesize the sample are hidden and randomized. The listener can listen to each test sample as many times as desired. The listener is asked to eliminate the worst of the available samples sequentially. Only the sample being heard can be eliminated. Once a sample is eliminated, its rank is displayed on the screen. The elimination of the samples can be stopped if the remaining samples are perceptually similar. This results in such remaining samples having the same rank. A screenshot of the RBE test implemented by us is shown in Fig. 1. Our modification to the WebMUSHRA implementing this RBE test is available to the

public at https://s.fhg.de/rbe23.

Statistical analysis on the generated rankings can be done through the Plackett-Luce model [15]. This model produces a worth value for the TTS models on a logarithmic scale, which can then be compared. Further, the Plackett-Luce model also provides the p-values indicating the statistical significance of the difference between the TTS models.

## 3. Controlled Test Conditions

### 3.1. Text-to-Speech Model

We used ForwardTacotron [3] with the StyleMelGAN vocoder [16] as the TTS model under test. This low-complexity TTS model can achieve state-of-the-art performance faster than real-time inference speeds [17].

ForwardTacotron is a single-pass non-autoregressive model built with feedforward networks, gated recurrent units, and long short-term memory networks [17]. This model predicts mel-spectrogram frames from the corresponding input phoneme sequence. Further, a duration predictor predicts the duration of individual phonemes to which the length regulator resamples the predicted mel-spectrogram frames. Finally, the mel-spectrogram frames are transformed into audio frames using StyleMelGAN, which is a generative adversarial network-based neural vocoder built using temporal adaptive denormalization blocks.

### 3.2. Test Sentences and Preliminary Objective Evaluation

Zielinski et al. [10] reported that indirect scaling tests like RBE are more accessible than direct scaling tests like ACR. In indirect scaling tests, the listener is only asked to rate if one sample is better. It would be interesting to know whether the evaluation results from these two methods differ. Further, it would be interesting to know if the ratings depend on the presence or ab-

Table 1: *Training losses at each of the selected checkpoints with MCD-DTW, MSD-DTW, and f0RMSE-DTW in relation to the 500K step trained model. Only the mean of the objective metrics over 40 Harvard sentences is shown.*

| Training checkpoint (K iterations) | Training Loss ($\downarrow$) | MCD-DTW ($\downarrow$) | MSD-DTW ($\downarrow$) | f0RMSE-DTW ($\downarrow$) |
|---|---|---|---|---|
| 5 | 1.382 | 17.84 | 32.20 | 238.87 |
| 25 | 1.088 | 15.60 | 27.96 | 209.68 |
| 50 | 0.960 | 14.40 | 26.16 | 198.01 |
| 140 | 0.930 | 11.79 | 21.42 | 167.49 |
| 500 | 0.725 | – | – | – |

sence of perceptually close samples. Hence, we performed the subjective evaluations with the following two test sets:

1. Perceptually distinguishable set (PDS): Here, outputs from all the TTS models being evaluated are clearly perceptually distinct from each other.

2. Perceptually indistinguishable models in set (PIS): There is at least one TTS model whose output is perceptually very close to the output from the best TTS model in informal listening.

To simulate the TTS models with varying degrees of perceptual quality, we used a single-speaker training of the ForwardTacotron model at various stages of training. We used the LJ Speech [18] as the dataset for training the ForwardTacotron model. We held out Chapter 50 for testing, with the remaining 49 chapters used for training. This resulted in approximately 23 hours of training data. Out of the training chapters, ten samples at random were left for validation. These validation samples were used for informal hearing tests at intermediate checkpoints during the training using TensorBoard logging. The training loss, which correlated with the validation loss, was used as an indicator to select the checkpoints for the subjective analysis.

The output was intelligible at low quality for a model checkpoint after 5K training iterations with a batch size of 32. Therefore this model checkpoint after 5K iterations was used as one of our models under test. The last training checkpoint at 500K iterations was identified to be the best model. We further chose model checkpoints at different training iterations where the synthesis outputs were perceptually different from one another.

We verified that chosen checkpoints produced perceptually different quality speech using the objective metrics of mel-cepstral distortion (MCD), mel-spectrogram distortion (MSD), and f0 root mean square error (f0RMSE). The objective metrics indicated that these could be different from one another. Further, through informal listening, we verified that there are easily perceptible differences between the outputs at the chosen training checkpoints.

As multiple definitions for the objective metrics exist in the literature, we present those used in our study here for completeness. The MCD was computed using [19]

$$\text{MCD}_{XY} = \frac{1}{N} \sum_{n=1}^{N} \sqrt{\sum_{k=1}^{K} (\text{MC}_X(k,n) - \text{MC}_Y(k,n))^2}, \quad (1)$$

where $\text{MC}_X(k,n)$ and $\text{MC}_Y(k,n)$ represent the $k$-th cepstral coefficient in $n$-th time frame for the file $X$ and $Y$, respectively. The mel-cepstral values were in decibels (dB). The MSD was

computed using

$$\text{MSD}_{XY} = \frac{1}{N} \sum_{n=1}^{N} \sqrt{\sum_{k=1}^{K} (\text{MS}_X(k,n) - \text{MS}_Y(k,n))^2}, \quad (2)$$

where $\text{MS}_X(k,n)$ and $\text{MS}_Y(k,n)$ represent the $k$-th spectral coefficient in $n$-th frame for the files $X$ and $Y$ respectively. We converted the spectral coefficients to the decibel (dB) scale before using them in this formula to get the MSD values in dB. Finally, the f0RMSE was computed using [20]

$$\text{f0RMSE}_{XY}$$
$$= 1200 \sqrt{\frac{1}{N} \sum_{n=1}^{N} (\log_2(F_X(n)) - \log_2(F_Y(n)))^2}, \quad (3)$$

where $F_X$ and $F_Y$ represent the f0 values for the file $X$ and $Y$, respectively . The resulting f0RMSE is given in cents.

We used our best model, the model checkpoint trained up to 500K iterations, as our reference model for calculating the objective metrics. Since the durations of the synthesized samples vary across checkpoints, we used dynamic time warping (DTW) to align the samples. The DTW cost for the MCD-DTW, and MSD-DTW was the mel-cepstral and mel-spectral distance, respectively. To calculate the f0RMSE-DTW, we aligned time frames across files by finding the minimum MCD cost path between them. This allowed us to calculate the squared f0 error across all the frames even though meaningful f0 values existed only for those consisting of voiced phonemes. The f0 values for f0RMSE were measured using the SWIPE algorithm from libf0 [21]. These values were used in f0RMSE calculation only if the prediction confidence for each of the corresponding frames was greater than 0.4.

We used the training checkpoints starting from barely intelligible (after 5K training iterations at a batch size of 32) to close to the best possible quality (after 140K training iterations). The selected checkpoints, their training losses, and the objective metrics are shown in Table 1. Here the objective metrics are averaged over the first 4 lists of Harvard sentences consisting of 40 sentences [22].

## 4. Results and Discussion

We conducted the subjective tests using 40 sentences from the first 4 Harvard sets with 16 listeners each. Each listener listened to outputs from all the TTS models under test (4 with PDS and 5 with PIS case). We limited each listener to listen to 20 sentences per TTS model. This meant, in total, we had 320 ratings per TTS model. We used the WebMUSHRA software for this purpose and utilized listeners from the lab not connected

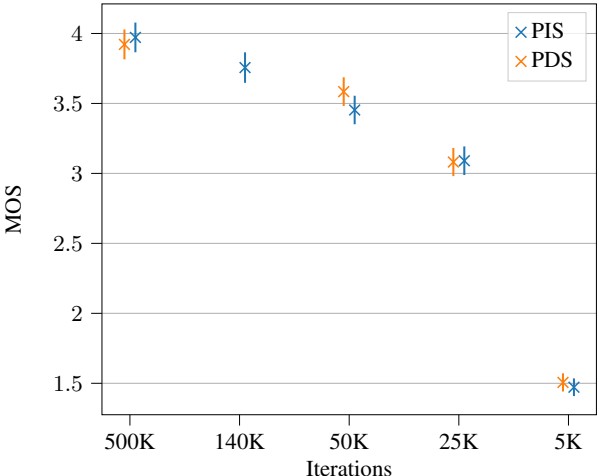

Figure 2: *MOS values with their confidence intervals across model checkpoints for the perceptually distinguishable set (PDS) and perceptually indistinguishable set (PIS).*

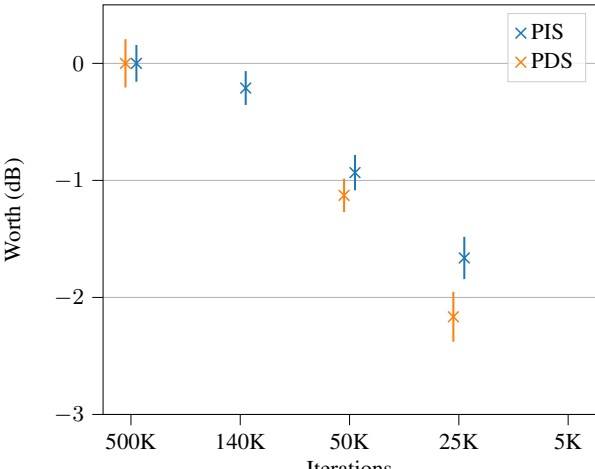

Figure 3: *Worth values with their confidence intervals across model checkpoints for the perceptually distinguishable set (PDS) and perceptually indistinguishable sets (PIS) measured with the RBE test. The Worth values for the checkpoint at 5K iterations are $-10.6 \pm 2.66$ dB and $-6.3 \pm 0.96$ dB, respectively, and hence outside the plot range.*

Table 2: *Age and gender distribution across the different tests.*

| Test | Average age (y) | Males | Females |
|---|---|---|---|
| PDS-ACR | 31.8 | 9 | 7 |
| PDS-RBE | 31.4 | 11 | 5 |
| PIS-ACR | 29.3 | 11 | 5 |
| PIS-RBE | 31.5 | 13 | 3 |

Figure 4: *Box plot showing the time taken for each of the tests across different subjective test sessions.*

to the project. The age and gender distribution of the listeners varied across tests, shown in Table 2. All the listeners in the tests were fluent speakers, not necessarily native speakers, of English. From the ACR test results, we calculated the average and 95% confidence interval, and checked if the models under test were statistically significantly different using the Wilcoxon-signed-rank test. From the RBE test results, we calculated the worth values using the Plackett-Luce library from R, which also provided the p-values to determine the statistical significance of the differences.

The ACR and RBE test results for both of our two test sets, PDS and PIS, are shown in Figs. 2 and 3, respectively. The PIS and PDS checkpoints are plotted in both plots with an offset for better visibility. As determined by both tests, the preference order of the TTS models was identical, as we can see from the results. Both tests across the two scenarios also showed that each pair of successive checkpoints (from the chosen ones) was statistically significantly different; the p-values are shown in Table 3.

If we look at the two tests done using ACR, the resultant intervals between the scores of consecutively selected checkpoints remain approximately the same for PDS and PIS. A similar conclusion can be drawn for the RBE results. Further, from the RBE test results, we note that the calculated worth values for the lower-quality checkpoints, i.e., 50K and lower, are al-

ways lower for the PDS than for the PIS. This indicates that it was easier to eliminate the samples for the PDS. Hence, more correct decisions in terms of quality were made when there was a clear perceptible difference between samples.

All four tests showed that the used TTS models were all statistically significantly distinct, as shown in Table 3. We also did a significance analysis using smaller subsets of the listeners for both tests. For the most challenging case of 140K vs. 500K comparison, more than 12 listeners were required to get the significance values distinct for both tests. Hence, the RBE test also requires a similar number of participants as the ACR test for reliable results.

We also recorded the time the listeners took to perform the ratings, which are shown in Fig. 4. We can conclude that the RBE tests take longer than the ACR test, almost double in the median sense. Even so, the overall median time to rank the five samples in each rating is 40 s, which may be acceptable for usage. For the ACR test, the time taken does not depend on the test set, meaning the time required to rate each sample remains

Table 3: *p-values for statistical significance, with a value less than 0.05 indicating statistically significant differences.*

| Checkpoints | ACR Test | | RBE Test | |
| --- | --- | --- | --- | --- |
| | PDS | PIS | PDS | PIS |
| 500K vs. 140K | - | 1.7e-5 | - | 0.04 |
| 500K vs. 50K | 1.3e-9 | - | 2e-16 | - |
| 140K vs. 50K | - | 1.9e-8 | - | 1.6e-10 |
| 50K vs. 25K | 1.7e-18 | 1.3e-10 | 2e-16 | 1.5e-10 |
| 25K vs. 5K | 3.2e-54 | 1.3e-51 | 8.9e-6 | 2e-16 |

almost the same across both PDS and PIS scenarios. This is expected since the samples are rated independently. We can also see that for the RBE test, the time taken to rate the individual items of the PIS test (median of 9.5 s per ranking item) is higher than the PDS test (median of 7.2 s ). This intuitively makes sense since the listener must listen more carefully to the samples when some are perceptually very close.

We conducted additional tests using different checkpoints (i.e., after 150K and 200K training iterations) instead of the 140K checkpoint. The ACR and RBE tests showed no significant differences between the best two checkpoints (i.e., 500K vs. 200K or 500K vs. 150K). While these results do not provide new information, they support our finding that the RBE and ACR tests yield similar results.

We did not use the "held out" Chapter 50 of the LJSpeech database for listening tests due to the presence of a lot of "seen" words. The LJSpeech dataset's textual content generally consists of many law-enforcement terminologies. Even in the unseen Chapter 50, the sentences had a lot of seen terms like "prison," "president," "commission," etc.

## 5. Conclusion

In this paper, the ACR test was evaluated for the ability to distinguish perceptually similar outputs from various TTS models. We investigated whether an indirect scaling based multi-stimulus RBE test provides similar results. In the RBE test, worst quality samples are eliminated one after the other in a loop to get a ranking for the samples. These rankings were then analyzed using a Plackett-Luce model, which assigns a worth value to each TTS model and calculates the statistical significance measure. We also extended the WebMUSHRA framework to support the RBE test.

To simulate TTS models with different perceptual qualities, we used model checkpoints from various iterations while training a single-speaker non-autoregressive ForwardTacotron model. We conducted subjective evaluations using ACR and RBE tests under distinct and indistinguishable scenarios. Our results showed that ACR test results are comparable to RBE and that it was possible to differentiate between perceptually close models using both tests. Since indirect scaling tests such as RBE are easier to perform than ACR tests, they could be used in cloud-based TTS evaluations.

## 6. Acknowledgements

Parts of this work have been supported by the SPEAKER project (FKZ 01MK20011A), funded by the German Federal Ministry for Economic Affairs and Climate Action. In addition, this work was supported by the Free State of Bavaria in the DSAI project.

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
