# OpenReview forum: "Subjective Evaluation of Text-to-Speech Models: Comparing Absolute Category Rating and Ranking by Elimination Tests"
_Interspeech.org/2023/Workshop/SSW — SSW12_

### Official Review · Reviewer_H8Sc · 2023-06-02
**A comparison of Absolute Category Rating and Ranking By Elimination type listening tests for perceptually-distinguishable and perceptually-indistinguishable samples**

**Rating:** 4
**Confidence:** 4

**Review:**

This paper presents a comparison of two types of listening tests: the Absolute Category Rating test (basically, MOS) and a Ranking By Elimination (RBE) test wherein listeners eliminate the worst sample in a set one by one.  Furthermore, two types of test sets are considered: one wherein all the systems are "perceptually different," and another set that is "perceptually indistinguishable."  It's nice to see a direct comparison of two listening test methodologies like this, especially with a focus on how they each can serve to distinguish close-in-quality vs. clearly-distinct test sets.  The different quality samples were created by using different training checkpoints of a ForwardTacotron model trained on LJSpeech.  Looking at the time taken to complete each type of test is interesting and informative, and extending WebMUSHRA to support RBE is a nice contribution.

One main issue with this paper is that I am not really sure what the takeaway message was.  It looks like ACR tests can be completed by listeners more quickly, and both tests have basically the same ability to distinguish between perceptually-similar and perceptually-different test sets.  Is there any scenario where it would be beneficial to use RBE instead of ACR, or is ACR the clear winner?  Are there any recommendations for when to use one type of test or the other?

The other main issue with this paper is that the perceptually-indistinguishable test set (PIS) does not seem to actually well-represent the trait of being perceptually indistinguishable -- in particular, the PIS is defined as "There is at least one condition that is perceptually very close to the best condition in informal listening."  From the name (perceptually indistinguishable *set*), as well as from the motivation described for using it, it sounds like the entire set should be perceptually close, not just the top two systems.  Having the entire set be perceptually close would make for much stronger claims.  The PDS is generated from model training checkpoints 5, 25, 50, 140, and 500, and the PIS appears to be simply exactly the same set minus the model from checkpoint 140.  This also introduces a confounding factor that the different test sets have a different number of conditions, and that claims such as "for the RBE test, the rating for the PIS takes more time than the PDS" may trivially be explained by the fact that there are simply fewer conditions to listen to in the PDS test.

There was also confusing and contradictory information about the data used in the experiments, in particular, the test set.  Section 3.2 states that Chapter 50 of the LJSpeech data was held out for testing.  But then, the objective metrics were computed on synthesized Harvard sentences, and the listening test was also conducted using the Harvard sentences -- it was not clear how and whether LJ Chapter 50 was used at all.  It also wasn't clear why Harvard sentences with no ground-truth audio were chosen especially when objective measures were being used.  Synthesis from the best model was used as the reference for the objective metrics, which meant that the best model could not be objectively evaluated at all -- why not use Chapter 50 for these objective metrics and compare with ground-truth audio?  Likewise, it was stated that "Due to the small validation subset, the training loss was used as an indicator to select the checkpoints for the subjective analysis."  Then, why were only 10 validation samples chosen to be held out?  What were the validation samples used for, if not for model selection?

There are several more minor issues and points needing clarification:
* I would like to have seen a (short, e.g. half a sentence) description of what an RBE test entails much earlier on in the paper, e.g. the first time it is mentioned in the introduction.
* In Section 3.2: "It is commonly assumed that single stimulus tests, such as the ACR test, do not work well when the conditions under test are perceptually very close."  Assumed by whom?  In fact, the opposite is known to be true -- a bias known as "range-equalizing bias" describes that listeners use the full range of category rating options, regardless of the overall range of quality of samples presented (described here:  http://www.acourate.com/Download/BiasesInModernAudioQualityListeningTests.pdf )
* The meaning of "conditions" is occasionally unclear -- sometimes it is used to mean TTS systems, other times it seems to refer to individual samples.
* Likewise, the words "iteration" and "checkpoint" are being used interchangeably, which was initially confusing.
* Age and gender distributions are shown for the listeners -- it also would be nice to see some info about whether or not they are native listeners of English.
* What is "a castanets file"?
* In the conclusion: "We conducted a study to determine how well the ACR test can evaluate perceptually similar outputs from various TTS models.  To do this, we used a multi-stimulus RBE test where progressively worse quality conditions are eliminated."  These sentences do not match the content of the paper at all -- weren't both types of tests conducted to evaluate how well both types of tests can evaluate perceptually similar samples?  Isn't the worst sample eliminated first, so progressively better and better samples are being eliminated one by one?

Some additional small points related to presentation:
* Figure 1: The orange rectangle in the middle of the figure seems to be a lot of empty space.  What is this supposed to show?  Furthermore, the text (which is actually very helpful for understanding the RBE test) is very small and difficult to read.
* In Figures 2 and 3, it looks like the PIS and PDS chosen checkpoints are actually slightly different (e.g., at 25k, it looks like the PDS one is slightly less than 25k and the PIS one is slightly more) -- is this in fact the case?  If so, why?  If not, then why is it shown this way?

---

### Official Review · Reviewer_eskF · 2023-06-05
**Interesting work but the experimental validation is limited**

**Rating:** 7
**Confidence:** 4

**Review:**

This paper presents a comparison between an ACR method and a ranking-by-elimination method in subjective evaluation of TTS systems. Evaluation sets were artificially generated by varying the number of iterations in neural TTS training. The experimental results showed that both methods yielded similar results.

This paper is relatively well written. Research motivation is clear. The research topic dealt with in this paper is interesting. This comparison is supposed to be novel. It is worthwhile to revisit subjective evaluation methods and try to find a better way to carefully evaluate synthetic speech to fairly compare various TTS systems.

One of the weak points of this paper is that an experimental design is limited. It will be required to conduct more experiments in various conditions in order to make conclusions. For instance, the number of conditions was set to only 5. On the other hand, more conditions, i.e., more systems or methods, are often evaluated in TTS research area, e.g., more than 20   systems need to be evaluated in recent challenge activities, such as Blizzard and VCC. It is expected that the number of conditions would significantly affect the results. I guess that some small differences that can be detected by a preference method would not be found by the ACR method as the number of conditions increases. It will be interesting to investigate whether or not the ranking-by-elimination method can do.

The other weak point is that only the ACR method and the ranking-by-elimination method were compared. As mentioned in Section 1, MUSHRA is similar to the ranking-by-elimination in terms of multiple samples can be compared in scoring. Therefore, it is worthwhile to conduct comparisons among more evaluation methods including MUSHRA, a preference test, and so on, and clarify pros and cons of each method.

---

### Decision · Program_Chairs · 2023-06-14

**Decision:**

Accept

**Comment:**

SSW2003 received 45 papers. The acceptance rate is 82%. We are pleased to inform you that your paper has been accepted by the SSW2023 Program Committee. Please read the reviews carefully and submit your camera-ready paper by June 28th. Most of reviewers performed a detailed review. Please answer to their questions and take into account their comments.
Since your paper received a score below 5/9 that is strongly argued by the reviewers, note that the Program Committee will check if your manuscript has been significantly changed to specifically consider their remarks. Note that camera-ready papers are credited with one extra page to allow authors to consider reviewers’ suggestions. So max 7 pages in total including figures & refs.
The deadline for submitting the revised version (with full non anonymized authors and refs!) is 28th June.